# Key Process and Factors Controlling the Direct Translocation of Cell-Penetrating Peptide through Bio-Membrane

**DOI:** 10.3390/ijms21155466

**Published:** 2020-07-30

**Authors:** Kazutami Sakamoto, Taku Morishita, Kenichi Aburai, Kenichi Sakai, Masahiko Abe, Ikuhiko Nakase, Shiroh Futaki, Hideki Sakai

**Affiliations:** 1Department of Pure and Applied Chemistry, Faculty of Science and Technology, Tokyo University of Science, 2641 Yamazaki, Noda, Chiba 278-8510, Japan; taku.m0713@gmail.com (T.M.); ken.aburai@gmail.com (K.A.); k-sakai@rs.tus.ac.jp (K.S.); abemasa@rs.tus.ac.jp (M.A.); hisakai@rs.tus.ac.jp (H.S.); 2Nanoscience and Nanotechnology Research Center, Research Organization for the 21st Century, Osaka Prefecture University, Naka-ku, Sakai, Osaka 599-8570, Japan; i-nakase@21c.osakafu-u.ac.jp; 3Institute for Chemical Research, Kyoto University, Uji, Kyoto 611-0011, Japan; futaki@scl.kyoto-u.ac.jp

**Keywords:** cell-penetrating peptide (CPP), direct permeation (cytolysis), giant unilamellar vesicle (GUV), CPP adsorption, FITC-octa arginine (FITC-R8)

## Abstract

Cell-penetrating peptide (CPP) can directly penetrate the cytosol (cytolysis) and is expected to be a potent vector for a drug delivery system (DDS). Although there is general agreement that CPP cytolysis is related to dynamic membrane deformation, a distinctive process has yet to be established. Here, we report the key process and factors controlling CPP cytolysis. To elucidate the task, we have introduced trypsin digestion of adsorbed CPP onto giant unilamellar vesicle (GUV) to quantify the adsorption and internalization (cytolysis) separately. Also, the time-course analysis was introduced for the geometric calculation of adsorption and internalization amount per lipid molecule consisting of GUV. As a result, we found that adsorption and internalization assumed to occur successively by CPP molecule come into contact with membrane lipid. Adsorption is quick to saturate within 10 min, while cytolysis of each CPP on the membrane follows successively. After adsorption is saturated, cytolysis proceeds further linearly by time with a different rate constant that is dependent on the osmotic pressure. We also found that temperature and lipid composition influence cytolysis by modulating lipid mobility. The electrolyte in the outer media is also affected as a chemical mediator to control CPP cytolysis by following the Hoffmeister effect for membrane hydration. These results confirmed the mechanism of cytolysis as temporal and local phase transfer of membrane lipid from Lα to Mesh_1_, which has punctured bilayer morphologies.

## 1. Introduction

Cells are the basic unit of life separated from the environment by a lipid membrane, and life is a system in which molecules come in and out while the system keeps its structures and functions consistent under homeostatic control. While the water-soluble peptide called the cell-penetrating peptide (CPP) has been known for its spontaneous active penetration through bio-membranes without interfering with cell viability and is expected to be a potent vector for a drug delivery system (DDS) to carry bioactive cargo [1,2,3]. There are two known ways of internalization of CPP—one is endocytosis, by which CPP is entrapped in the endosome into the cytosol, and the other is cytolysis as a direct translocation of CPP through the lipid membrane [1,2]. There are several pathways of endocytosis reported, such as actin-driven micropinocytosis (diameter of endosome is >1 µm), clathline mediated endocytosis (~120 nm), and caveolae endocytosis (~80 nm), each of which may give a choice of CPP and cargo size [1,2,3,4,5,6]. By endosomal internalization, CPP must be released from endosome to cytosol; in this regard, cytolysis is a direct and efficient way to deliver bioactive cargo to the target [1].

HIV-1 Tat (48–69) is the most well-known CPP among the many natural and synthetic CPPS with which the above-mentioned different pathways, i.e., endocytosis and cytolysis, have been revealed through extensive studies with varieties of cultured cell systems. Futaki et al. reported that key element of the CPP’s cellular uptake is the richness of cationic amino acids especially arginine in the peptide molecule [2]. They found that the oligomer of Arg, such as R8 and R9, behaves just as the TAT peptide for various cultured cells and proposed the existence of a common internalization mechanism [3]. Wender et al. further generalized the basic unit as the number and array of guanidium groups even for the synthetic oligomers such as a peptoid molecule [4].

In terms of the clarification of the pass way and mechanism of CPP translocation to the cytosol, it was quite difficult to differentiate unit processes such as adsorption to internalization and endocytosis to cytolysis in a cultured cell experiment. Richard et al. reported the way to separate the adsorption and internalization of CPP by applying trypsin to digest adsorbed CPP on the outer cell membrane for TAT and R9 cultured with HeLa and Jurkat cells. They have shown that temperature-dependent internalization occurred by endocytosis for both TAT and R9, but no details were examined for cytolysis [7]. As such, there are examples of CPP cytolysis, and ways to translocate have been reported, although the conclusive mechanism is yet to be established because of the difficulty of separating each process that concurrently occurs under cultured cell experiments [1]. Our objective is to make a clear understanding of cytolysis and provide a way to control CPP’s direct internalization. In order to separate adsorption to internalization and cytolysis to endocytosis, we have used FITC labeled R8 (FITC-R8) as a model CPP and giant unilamellar vesicle (GUV) as a cell model along with trypsin digestion.

We have taken the cytolysis of CPP as a physicochemical phenomenon of the lipid membrane common to the structure and phase changes of lyotropic self-assembly of amphiphiles, as shown in Appendix A [8,9]. Previously, we have proposed a specific mechanism for cytolysis of CPP as a temporal and local phase transition from Lα to Mesh_1_, which has punctured bilayer morphologies, caused by electrostatic adsorption of cationic guanidino groups of oligoarginine in the CPP to generate positive curvature at the point of absorption to the phosphate anion of membrane lipid, as shown in Appendix A [10,11,12,13]. Hyde et al. reviewed the existence of intermediate mesophases, including the Mesh_1_ phase, and explained their importance to the functions of the bio-membrane [14]. Recently, we have shown that, under the equilibrium condition, the incremental addition of oligoarginine to lyotropic liquid crystals, composed of phospholipids in water, changed their self-assembly structure from Lα lamellar to hexagonal (H_1_) via bi-continuous (V_1_) by increasing the positive mean curvature. During that course, the definite existence of Mesh_1_ phase was confirmed just as expected from Hyde’s theory [14]. This is a sufficient indication that absorption of CPP would induce local and temporal dynamic topological deformation of the membrane structure to Mesh_1_, which has pores in the lamellar and inspired the possibility of controlling CPP cytolysis by combining other factors that are effective in changing the membrane curvature [10,11,12,13]. Taking this into account, we used osmotic pressure as a physical mediator to change the membrane curvature using fluorescence isothiocyanate labeled octa-arginine (FITC-R8) as a CPP. We have measured the cytolysis of CPP by changing the osmotic pressure of the outer media for GUV and erythrocyte. As expected for our proposed mechanism, the reduction of osmotic pressure, which induced positive membrane curvature, enhanced cytolysis for both GUV and erythrocyte because CPC needs less work for the deformation of membrane morphology [10,11,12]. Thus, we have confirmed the credibility of the proposed mechanism and showed a promising way to control CPP cytolysis by changing the membrane curvature through physical modulation. For these experiments, we analyzed the FITC-R8 amount in the GUV or erythrocyte by high performance liquid chromatography (HPLC) [10,13]. Total recovery of FITC-R8 as a sum of cytolysis and supernatant after centrifuging was over 92% under varying osmotic pressure changes, which confirmed the reliability of the experiment [15].

Recently, Yamazaki et al. reported the amount of CPC translocated to the GUV with carboxyfluorescein labeled Nona-arginine (CF-R9) as the CPP by analyzing the fluorescence intensity with a double GUV image, a small vesicle in the GUV, obtained by confocal laser scanning microscopy (CLSM) [16]. They measured the fluorescence intensity of CF-R9 adsorbed on the rim of GUV quantitively with a time course that corresponds to the CF-R9 adsorption on the outer membrane surface. They also observed the internalized CF-R9 on the rim of the small vesicle inside GUV qualitatively. As a result, they concluded that there are two modes of entry of CF-R9 into GUV. One is the mode A internalization by which CF-R9 internalize directly across the lipid membrane without the leakage of fluorescent dye encapsulated in the GUV. They named this mode as prepore internalization, and the amount of internalized CF-R9 is quite low. While by the mode B at which CF-R9 enters through the pore, the rate and amount of CF-R9 entry were much higher than the mode A, and leakage of fluorescent dye through the pore was observed. They called the mode A as translocation through prepores [16]. They reported other mode A type internalization for CF-TP10, truncated analog of transportan with fluorescent prove [17]. Based on these findings, Yamazaki et al. suggested the hypothesis to describe the mechanism of CPP translocation—namely, that adsorbed CPP on the GUV surface affects the molecular interaction between lipids, which induce prepore (mode A) or via pore (mode B) CPP internalizations [16,17]. Almeida et al. also reported similar translocation without pore formation for another variant of TP-10 with fluorescent Rhodamine label (Rh-TP-10W) and called it silent translocation [18]. They explained such membrane permeabilization to be the results of peptide-induced stochastic fluctuations in lipid organization in the bilayer. Both results and the explanation by Yamazaki and Almeida for the prepore (or silent) internalization of CPP is in good agreement with our proposal of the cytolysis translocation mechanism and the effect of curvature modulation to control the level of cytolysis. We also concluded that cytolysis of CPP and the internalization of the anti-microbial peptide (AMP) through the pore falls into the same physicochemical transformation of bilayers that can be explained as a general phenomenon in membrane shape deformations such as the area difference elasticity theory [13,19,20,21].

As mentioned above, there is general agreement for the mechanism of CPP cytolysis as internalization by the dynamic process of membrane deformation, even though there are no distinctive analyses to clarify the CPP adsorption and actual internalization mechanism (whether they are simultaneous or successive phenomena). Here, we report the key process and factors controlling the direct translocation (cytolysis) of CPP through bio-membrane. To elucidate the task, we have introduced trypsin digestion of adsorbed CPP onto GUV to quantify the adsorption and internalization (cytolysis) separately. Also, the time-course analysis was introduced for the geometric calculation of adsorption and cytolysis amount per lipid molecule consisting of GUV. As a result, we found that adsorption and cytolysis are assumed to occur successively by each single CPC that come into contact with membrane lipids. Adsorption is quick to saturate ca. within 10 min for 100 µM FITC-R8. While cytolysis seems slow at the initial stage because adsorption is rate-limiting, after adsorption saturation, it follows a linear time course, and the rate constant is dependent on the osmotic pressure (it reflects the curvature of the membrane). We also found that temperature and lipid composition influence the internalization by means of modulating mobility of lipids, i.e., the flexibility of the membrane to fluctuate. The electrolyte in the outer media also effected as a chemical mediator to control CPC cytolysis by following the Hoffmeister effect for the hydration of the membrane surface.

## 2. Results

### 2.1. Characterization of GUV Geometry

The translocation of CPP to GUV reported in the past were total amounts of CPP trapped by GUV and there is no clarification of adsorption and cytolysis [10,11,12,13,15,16,18]. In order to differentiate these two amounts, we have introduced trypsin digestion of GUV after incubation with CPP as shown in Section 4.2 Methods and Appendix A. The actual cytolysis amount was obtained from the trypsin treated GUV by Triton X-100 (TX100, Sigma-Aldrich Japan, Tokyo, Japan) solubilization of lipids and then fluorescent intensity measurement for FITC-R8. Previously, we have shown the CPP translocation to GUV as the amount (µM) detected vs. applied (usually 100 µM) with a satisfactory high recovery rate (over 90%) for each experiment [15]. On the other hand, the process applied here through second incubation with trypsin reduced the recovery rate by about 60%. To secure the reliability of data and get an insight of actual CPP translocation, we have calculated the amount of adsorption and cytolysis as µM of CPP/mol of membrane lipid composing the outer layer of GUV by applying the geometrical parameter of GUV calculated with the data shown in Table 1 and Appendix A. Previously, we have reported that CPP translocation occurs by electrostatic adsorption of CPP to generate positive curvature at the point of absorption, which then cause temporal and local phase transition from Lα to Mesh_1_ (Appendix A) [10,11,12,13,15]. In this regard, expression of the amount of adsorption and cytolysis by µM of CPP/mol of lipid in the outer layer of GUV gives us a practical image of how each CPP molecule acts with GUV.

GUV formation was confirmed by differential interference contrast microscope (DICM), and particle size distribution was measured by dynamic light scattering (DLS), as shown in Figure 1 for EPC GUV under the isotonic condition at 56 mOsm. An average diameter (d) of 1.4 × 10^4^ nm was used for the calculation (Appendix A) of parameters shown in Table 1. Molecular occupation area (a) of egg yolk phosphatidylcholine (EPC) and 1-Stearoyl-2-oleoyl-phosphatidylcholine (SOPC) are measured by the surface pressure (π)–molecular occupation area (A) method as shown in Figure 2. When the curvatures of surface pressure (π)–molecular occupied area (A) were extrapolated to surface pressure = 0, the ultimate molecular occupation area per one molecule of phospholipid (a) can be calculated. From Figure 2, the ultimate molecular occupation area of EPC and SOPC were calculated as 0.71 nm^2^/molecule and 0.84 nm^2^/molecule, respectively, which correspond well to the reported results [22,23].

### 2.2. Cytolysis of CPP to GUV

#### 2.2.1. Distinctive Analysis of Cytolysis to Clarify the CPP Adsorption and Actual Internalization

As mentioned in the introduction, the amount of internalization of CPP to GUV reported was for the sum of adsorption and internalized CPP by cytolysis because of the experimental difficulty to quantify a small amount of CPP in the cytosol, especially from the fluorescence intensity of cytosol by CLSM image analysis of GUVs. In this study, we applied trypsin digestion after 10 min of incubation for translocation study to distinguish these two processes. As shown in Figure 3, the effect of osmotic pressure, which corresponds to the curvature of the lipid membrane, to the total amount of translocated FITC-R8 to SOPC GUV, including adsorption and cytolysis, which correspond to the data we have previously reported [8,9,10]. By reducing outer media osmotic pressure, which corresponds to the increasing membrane curvature toward the outside, the amount of translocation of FITC-R8 was increased. Here, 56 mOsm is the isotonic condition as 28 mM PBS for both outer and internal media of GUV.

Cytolysis amount was obtained after trypsin digestion [7,24] of GUV, and adsorption amount was calculated by subtracting cytolysis (C) from total translocation amount (T) for each osmotic pressure. Completion of trypsin digestion was checked by thin layer chromatography (TLC) analysis for the supernatants and no spot for the residual FITC-R8 was found. (TLC chromatogram is not shown here.) As shown in Figure 3, the amount of adsorption (A) seems to be independent from the osmotic pressure and surface potential of GUV as shown in Table 2. Zeta potential was measured for the vesicle with 1 µm diameter (1/14 of SOPC GUV) at which Zeta potential was more negative for hypertonic condition but became equivalent after adsorption of FITC-R8 as shown in Table 2. These results indicate that FITC-R8 with eight guanidium cations at the arginine sidechain would be more attracted to the membrane surface under lower osmotic pressure with more negative Zeta potential, but interestingly, the amount of adsorption was almost equivalent, and the resulting surface charge became the same level across the osmotic pressure range tested. These Zeta potential data confirm that adsorption and cytolysis are a successive but independent process.

#### 2.2.2. Time Course Analysis of CPP Translocation to GUV

Time dependence of total translocation (T) and cytolysis (C) of FITC-R8 to EPC GUV was measured by the method explained in Appendix A. The adsorption time course was obtained by subtracting C from T of each corresponding osmotic pressure and time by Equation (1).
Total translocation (T) = Adsorption (A) + Cytolysis (C)(1)

As shown in Figure 4, the adsorption of FITC-R8 was quite rapid and was already saturated after 10 min of incubation. Although there are some fluctuations, the amount of adsorption was almost similar in level regardless of osmotic pressure or time duration and tend to come to convergence after 60 min. On the other hand, cytolysis (Figure 5) starts rather slow in the beginning and then comes to a linear increase by time duration after adsorption reached saturation. The combined time course profile under the isotonic condition is shown in Figure 6. Those figures indicate that adsorption and internalization are assumed to occur successively for each single CPP molecule that comes into contact with the membrane lipid. The rate constant of cytolysis after 10 min was calculated from Figure 5 and shown in Table 3. It was found that the rate constant of FITC-R8 cytolysis to EPC GUV is faster for reduced osmotic pressure which corresponds to the amount of cytolysis profile to the osmotic pressure as shown in Figure 3 for SOPC GUV.

Yamazaki et al. [16] and Almeida et al. [18] reported similar CPC adsorption behavior to GUV. Yamazaki used carboxyfluorescein-labeled R9 (CF-R9) with DOPC GUV and showed CPP adsorption saturated within 50 s under the condition of 10 µM CF-R9 constantly micro-pipetted to the target GUV with 20 µ diameter. Almeida used two types of amphiphilic peptides TP10W and DL-1 with Rhodamine attached to N-terminal as a fluorescent probe (Rh-TP10W and Rh-DL-1a) as CPP with POPC GUV. They found the CPPadsorption saturated by 2–8 s for LUV (d = ca. 100 nm) under the condition of 1 µM of CPP to 50 µM of lipid (L/P = 50). These data by Yamazaki and Almeida are arbitrary units, but time resolutions are quite high because of the in situ observation of the fluorescence intensity of targeted single GUV under a microscope. While our method is to analyze whole GUV in each experiment under a given condition then sum-up to the time course data so that it is quantitative, although it is impossible to trace such rapid phenomena in seconds. All three trials are under good agreement that adsorption of CPP to the GUV surface is quick and saturates in seconds. Both groups, Yamazaki and Almeida, quantified the adsorption of CPP onto GUV from the fluorescence intensity of the rim of each single GUV image obtained by CLSM [17,18]. but they were unable to quantify the cytolysis because the fluorescence intensity of cytosol was too weak for quantitative analysis. Both of them utilized double vesicle GUV, and fluorescence of CPP adsorbed to the internal vesicle rim was used for the qualitative evidence of cytolysis. The behavior of independent GUV within the same microscopic image is not equal, and intensities are weak to quantify, perhaps because of the difficulty to focus on the location of the internal vesicle rim under cross-sectional CLSM analysis. As such, the method applied here as a successive combination of total translocation measurement and cytolysis measurement after trypsin digestion is a first quantitative analysis to distinguish adsorption and cytolysis separately for each experimental condition and to open up the possibility to investigate various factors effecting adsorption or cytolysis.

#### 2.2.3. Effect of Temperature for the CPP Translocation to GUV

Previously, we reported that translocation of CPP not only depends on the membrane curvature represented by osmotic pressure change but also on the mobility of the membrane lipid as shown by liquid crystal to gel phase transition [10,11,13]. We investigated the effect of temperature on the adsorption and cytolysis by the trypsin digestion procedure. As shown in Figure 7, under the gel phase, at which hydrophobic alkyl chain in the lipids is frozen to semi crystal, cytolysis of FITC-R8 almost ceased. On the other hand, adsorption to the GUV surface was not affected by temperature or osmotic pressure, as shown in Table 4. These results for cytolysis of FITC-R8 are in a good agreement with our proposed mechanism of temporal and local phase transition from Lα to Mesh_1_ because lipid molecules must be freely mobile under liquid crystal conditions to make structural rearrangement. On the other hand, adsorption is primarily electrostatic interaction of CPP to hydrophilic part of surface lipids as shown in Table 4, and this process only contributes to the amount of CPP available for cytolysis on the GUV surface.

#### 2.2.4. Effect of Lipid Composition for the CPP Translocation to GUV

The effect of cholesterol in addition to SOPC GUV for the total translocation of FITC-R8 was investigated. As shown in Figure 8, 30% replacement of cholesterol to the SOPC GUV suppressed the total translocation significantly. Although trypsin digestion was not applied, the level of translocation with 30% cholesterol is a similar level to the adsorption data shown in Table 4. As a result, it is assumed that cholesterol makes a membrane rather rigid (as is generally known) and suppresses the temporal and local phase transition to generate cytolysis. Yamazaki also reported similar suppression of cytolysis by cholesterol. They found no cytolysis (prepore internalization) for the CF-R9 to GUV composed of DOPG/DOPC/cholesterol (2/6/4), while GUV without cholesterol as DOPG/DOPC (2/8) showed cytolysis. The time course of CF-R9 adsorption was similar, but the amount of adsorption on the rim of GUV with cholesterol (33%) was about half of GUV without cholesterol. Their interpretation is that cholesterol increased the line tension of the membrane, which increases the necessary work for the membrane deformation like a flip flop or pore formation so that adsorbed CF-R9 stays only at the outer membrane layer. This is another way to explain the the rigidness of membrane lipid with cholesterol as discussed for our experiment.

#### 2.2.5. Effect of Solute in the Outer Media for the CPP Translocation to GUV

Previously, we proposed a way to control CPP cytolysis by maneuvering the membrane curvature [10,11,12,13]. There are two ways of controlling membrane curvature. One is physical modulation by changing osmotic pressure to induce surface pressure differences between the inside and outside of GUV as shown in Figure 3, Figure 7 and Figure 8. The other is chemical modulation by applying solute that is either kosmotropic (water structure makers) or chaotropic (water structure breakers) to influence the hydration level of membrane surface by the mechanism known as the Hoffmeister effect. We have successfully shown such effects for CPP (R8) translocation to living cells such as keratinocytes and fibroblasts [10,11,13]. The addition of NaSCN or 1,3-butanediol (both are chaotropic solutes) increased membrane curvature more positive and enhanced CPP translocation. On the other hand, sucrose as the kosmotropic solute suppressed the translocation.

Here, we have tried the Hoffmeister effect of solute on the translocation of CPP to GUV as a single cell model. By fixing Na^+^ as a cation, SCN^−^, Cl^−^ and SO_4_^2−^ were used as counter anions in the serial order based on the Hoffmeister effect. Namely, SCN^−^ is chaotropic, Cl^−^ is neutral, and SO_4_^2−^ is kosmotropic so that the order of positive curvature enhancement would be SCN^−^ > Cl^−^ > SO_4_^2−^. Experiments are conducted with varied solute concentrations. Figure 9 shows the result of FITC-R8 translocation (includes adsorption and cytolysis) to SOPC GUV and as expected SCN^−^ showed enhancement effect, while SO_4_^2−^ suppressed the translocation regardless of lipid/solute molar ratio.

## 3. Discussions

The cytolysis we are dealing with here is a direct internalization of CPP into cytosol without any residual pore in the membrane so that the barrier function and cell viability are intact during and after the CPP translocation. We have taken the cytolysis of CPP as a physicochemical phenomenon of the lipid membrane that is common to the structure and phase changes of lyotropic self-assembly of amphiphiles [10,11,12,13]. Previously, we proposed a specific mechanism for cytolysis of CPP as a temporal and local phase transition from Lα to Mesh_1_ caused by electrostatic adsorption of cationic CPP to the phosphate anion of membrane lipid in order to generate positive curvature, as shown in Appendix A [12,15]. We have confirmed the effect of CPP on the self-assembly structure of the membrane lipid under equilibrium conditions by the phase changes under the incremental addition of R6 (Hexa arginine as CPP) to the Lα LC composed of DOPC/Water (60/40 wt%) [15]. 

Such a phase transition corresponds to the surfactant parameter (*SP*) of the lipid molecule by increasing the cross-sectional area (*a*) at the hydrophilic group to generate positive membrane curvature. Here, *SP* is expressed by Equation (2). Islaelachvilli et al. proposed *SP* as the general relationship of the molecular geometry of amphiphiles to their self-assembly structure [8,9].
*SP* = *v*/(*a* × *l*)(2)
where *v* is the volume of hydrophobic parts of the lipid molecule, and *l* is the length of hydrophobic chains as shown in Appendix A [8,9]. *SP* is also related to the curvature of self-assembly by Equation (3) for surfactant monolayer.
*SP* = *v/*(*a* × *l*) = 1 + *Hl* + *Kl*^2^/3(3)
where *H* is the mean curvature and *K* is the Gaussian curvature [9]. This equation suggests that the shape of surfactant (membrane lipid) expressed as *SP* can be related to the global geometries of self-assembly, assuming homogeneous interfaces. The Lα type lamellar LC has zero mean curvature (*SP* = 1), and a slight modification of curvature toward positive (convexity toward aqueous phase) leads to bi-continuous cubic LC (V1:1/2 < *SP* < 2/3). Mesh_1_ is an intermediate phase between Lα and V_1_, which forms an array of holes with a catenoid structure as shown in Appendix A [14]. To make a curvature from planner surface (Lα) with *SP* = 1 to a hyperbolic curved surface, it would cost energy, which can be expressed as bending energy (*F_b_*) by Equation (4) [20].
*F_b_* = (*k_b_*_/_2) × (*H_b_*)2 + *Ҡ_b_* × *K_b_*(4)
where *k_b_* is the bending elastic constant, *H_b_* is the mean curvature, and *K_b_* is the bending Gaussian curvature as *K_b_* = 1/*C_b_*^2^. *Ҡ_b_* is the Gaussian curvature elastic modulus. As the Mesh_1_ phase appears between Lα and V_1_ phases, the energy cost to attain this modification would be quite small. If this pore formation were a local and temporal event, CPP, as a counter ion to the phospholipid, would translocate into cytosol without any distraction to the membrane integrity. This transition to Mesh_1_ occurs for R6 with DOPC at the molar ratio of DOPC/R6 as L/P (1/0.12 = 8.3/1) in equilibrium [15].

On the other hand, experiments reported here as CPP cytolysis shown in Figure 3, Figure 4, Figure 5, Figure 6, Figure 7, Figure 8 and Figure 9 were conducted with L/P (1/0.0012 = 850/1) for incubation, and the actual amounts of FITC-R8 (CPP) obtained for absorption or cytolysis on the GUV surface were less than 100 µmol (1/> 1 × 10^−5^) per 1 mol of EPC or SOPC, which are a far smaller in comparison to an equilibrium condition. By applying the a measured and geometrical data for GUV in Table 1, the distances between each FITC-R8 molecule on the EPC GUV surface are about 90 nm and occupied area by CPP per total GUV surface is <0.1% so that every FITC-R8 molecule was surrounded by lipids. Those results indicate that adsorption and successive cytolysis occurs for each single CPP molecule independently at the point of adsorption. This assumption supports the proposed mechanism of cytolysis as local and temporal phase change from Lα to Mesh_1_ by the dynamic process of membrane deformation.

Yamazaki called cytolysis prepore internalization caused by the binding of CF-R9 as CPP to the DOPG/DOPC GUV surface and inducing the tension in the membrane to stretch. They confirmed it by micro pipetting of 5–400 µM CF-R9 solution continuously at the vicinity of the GUV surface so that CF-R9 concentration was kept constant during adsorption and translocation. It is reasonable for their observation of prepore internalization (actually cytolysis) as the L/P ratio in this experiment is somehow close to our condition as the CF-R9 concentration was less than 400 µM. They also investigated the effect of surface charge density and lipid composition. Partial replacement of DOPC to anionic lipid DOPG enhanced CF-R9 adsorption to GUV and shorter hydrocarbon chain in lipid composition as DLPG/DTPC enhanced prepore internalization. In the latter case higher CPP concentration induced pore formation which was confirmed by the leakage of Fluorescent dye AF647 encapsulated in the GUV. This stable pore formation indicates the formation of Mesh_1_ domain in the GUV Lα membrane which resembles the generally accepted AMP’s pore formation [16]. Almeida et al. also reported similar behavior for amphiphilic AMPs as 0.75 µM of Rh-TP10W or Rh-DL-1a with POPC GUV (L/P = 1/0.05 = 67/1) at which internalization through pore reaches saturation about 20 min, but unexpectedly they found internalization without pore (cytolysis) of Rh-TP10W for some GUVs after 74 min. Although there is no explanation of the cause, they named it as silent internalization without significant perturbation to the GUV membrane [18]. We assume that the P/L ratio to the intact GUV was increased after long term incubation, which might be similar to the condition to our and Yamazaki’s for cytolysis experiments.

In terms of the factors effective to control the CPP cytolysis other than the osmotic pressure, we confirmed that temperature, lipid composition, electrolytes in the outer media influenced cytolysis as shown in Section 2.2.3, Section 2.2.4 and Section 2.2.5. Temperature and lipid composition are mainly relating to the mobility of membrane lipids as liquid crystal state which enables the membrane to respond bending energy (*Fb*) input to deform its shape with minimum energy level explained by the area difference elasticity theory [13,19,20,21,22]. As shown in Figure 7, SOPC is at the gel phase at 2 °C (T_c_ = 6 °C) so that lipid molecules are hydrated solid condition unable to rearrange their self-assembly structure. The addition of cholesterol suppressed cytolysis similarly to temperature change as shown in Figure 8 and also reported by Yamazaki. The amount of cholesterol in both experiments was coincidently similar to ca. 30%. They concluded that adsorption of CF-R9 to the outer surface of GUV is about the same level to the without-cholesterol but internalization is inhibited which is in good concordance with our data in Figure 8. Thus added cholesterol assumed to increased line tension of membrane which prevents the membrane deformation.

Regarding the effect of solute in the outside media, we showed the effect of electrolyte in the outer media as a chemical mediator to control CPC cytolysis in Figure 9. We found that the solute effect, whether chaotropic or kosmotropic by Hoffmeister definition, changed the level of cytolysis as a chemical modulator to control the cytolysis by affecting the hydration of the membrane surface as shown in Figure 9. This result supports the phenomena we have previously reported for the R8 translocation to the cultured cell, that was, NaSCN or 1,3-butanediol as chaotropic solutes enhanced CPP translocation. On the other hand, sucrose as a kosmotropic solute suppressed the translocation [10,11,15]. As such, all the factors explained here should be compared with the real biological systems such as cell cultures or even further to in vivo experiments in order to develop a practical way to control cytolysis for the various applications.

## 4. Materials and Methods

### 4.1. Materials

Egg yolk phosphatidylcholine (EPC) and 1-Stearoyl-2-oleoyl-phosphatidylcholine (SOPC) were gifted from NOF Co., Tokyo, Japan, and used for the preparation of GUV. Phosphate buffered saline (PBS, Wako Pure Chemical Co., Ltd., Osaka, Japan) was used as a solvent. FITC-R8 (fluorescein isothiocyanate was conjugated with γ -amino butyric acid as a linker to N-terminus of R8, synthesized at Futaki laboratory) was used as CPP. Triton-X100 (TX100, Sigma-Aldrich Japan, Tokyo, Japan) was used for the destruction of liposomes to examine the penetration ratio of FITC-R8.

### 4.2. Methods

#### 4.2.1. Preparation of GUV

GUVs were prepared by the freeze-melting and natural swelling method. Lipids dissolved in chloroform were placed into the vials, and then the solvent was evaporated. After being dried in vacuo for 16 h in the dark, lipid films were hydrated with 0.01 mg/mL of Rho/3 M KCl solution to give 20 mM lipid concentration. Lipid suspensions were sonicated for 30 min (CS-20, SHIBATA SCIENTIFIC TECHNOLOGY Ltd., Saitama, Japan), frozen using liquid nitrogen, and melted at room temperature. The suspension was vortexed for 1 min. This step (from freezing to vortexing) was performed six times. It was transferred to a dialysis membrane (UC-8-32-25; Viskase Co., Ltd., Chicago, IL, USA) and dialyzed against 20% PBS for at least two days at room temperature, and dialysis solution was daily replaced by a new one. In this study, 20% PBS wasdefined as isotonic solution and PBS solution at concentrations less than 20% as hypotonic solution and at higher than 20% as hypertonic solution.

#### 4.2.2. Confirmation of the GUV Formation

##### Differential Interference Contrast Microscope (DICM) Observations of GUV

ECLIPSE E-600 differential interference contrast microscope (DICM, Nikon, Tokyo, Japan) was used for the observations of the sample prepared by the method shown in Section 4.2.1 to confirm the formation of GUV.

##### Particle Size Distribution Measurement of GUV

A dynamic light scattering measuring apparatus (DLS, Nicomp 380ZLS, Agilent Technologies, Tokyo, Japan) using an argon laser (532 nm) was used for the Particle size distribution measurement of GUV.

#### 4.2.3. Phospholipid Concentration Measurement

The concentration of the phospholipids constituting the vesicles differs from the initial concentration by the dialysis process. Therefore, final concentration is adjusted by using a concentration measurement kit called c-test Wako. To 1.0 mL of C-test buffer, 20 μL of a 3.0 g/L BSA standard solution or vesicle suspension was added and incubated at 37 °C for 10 min. Thereafter, the adsorption of the standard solution or the sample was measured using an ultraviolet-visible spectrophotometer (wavelength: 600 nm).

The mol concentration of the phospholipid (M lipid) is represented by the following Equation (5).
M lipid (mol/L) = (3.0 (g/L) × A_Sa_)/(molecular weight of phospholipid (g/mol) × A_St_)(5)
where in A_Sa_ is the adsorption of GUV and A_St_ is the adsorption of the standard solution.

#### 4.2.4. Preparation of Trypsin Solution

In the course of membrane permeation, CPP first adsorbs to the vesicle surface. Therefore, to get the accurate permeation (internalization) amount, residual CPP adsorbed on the outer surface of the vesicle must be deducted. To do so, trypsin a basic amino acid-degrading enzyme was applied to the test solution after a penetration test to decompose CPP adsorbed on the membrane surface [22,23]. A method for preparing a trypsin solution is as follows; 5.0 g of trypsin was poured to 200 mL Erlenmeyer flask and 100 mL of 140 mM PBS (−) was added. After stirring the suspension for 2 h under a thermostatic chamber at 37 °C, the suspension was centrifuged (4 °C, 7000× *g*, 20 min) to remove insoluble residue, and then a 5% trypsin solution was prepared. Fractions of 10 mL were collected and stored as trypsin stock solution at −20 °C. For practical use, the trypsin stock solution was diluted 100 fold with PBS (−) to make a 0.05% trypsin solution that was used for the decomposition of CPP on the membrane surface.

#### 4.2.5. Surface Pressure (π)—Molecular Occupation Area (A) Measurement

In order to estimate the molecular occupation area of phospholipid consisting of a bilayer membrane, surface pressure (π)—molecular occupation area (A) measurement was applied. Here, 7.9 mg of SOPC and 10 mL of CHCl_3_ was added to a glass vial to prepare 1.0 mM of lipid solution. Then, 28 mM PBS (−), which is solvent, was filled in a trough of the surface pressure device, and 70 μL of the lipid solution was dropped by using a micro-syringe. After 15 min to let the CHCl_3_ completely vaporize, the surface pressure (π)—molecular occupation area (A) relationship was measured under 25 cm^2^/min of barrier pressure rate at 37 °C.

#### 4.2.6. Calculation of the Number of Phospholipid Molecules Consisting of GUV

Calculation of the number of phospholipid molecules consisting of GUV and other geometrical parameters are shown in the Appendix A. [25] The diameter of GUV is about 14 µm, N_tot_ = 1.7 × 10^9^, N_out_ = 8.8 × 10^8^ and N_int_ = 8.8 × 10^8^ are obtained.

The number of the vesicles N_vesic_ in 1 mL vesicle suspension is represented by N_vesi_ = (M_lipid_ × N_A_)/(N_tot_ × 1000), where in M_lipid_ and N_A_ represent molar concentration (mol/L) of the phospholipid and the Avogadro number (6.0 × 10^23^ (mol^−1^)), respectively. Since the molar concentration of the phospholipid is 1.7 mM, N_vesi_ is obtained as N_vesi_ = 5.82 × 10^8^ (unit/mL).

#### 4.2.7. Translocation Assay of CPP to GUV

The translocation amount (sum of adsorption and cytolysis) of CPP was measured by the following procedures (Appendix A). 600 μL of 20 mM GUV was centrifuged (4 °C, 10,500 rpm, 20 min) and washed three times with 20% PBS and 1200 μL of PBS at concentrations at 5%, 10%, 20%, 30%, and 40% (14, 28, 56, 84, and 112 mOsm) were added to precipitate and control the shape of GUV. Phospholipid concentration was adjusted to 10 mM each and lipid concentration was measured as shown in Section 2.2.3.

The following experiments were done to separate five liposomal suspensions. Here, 200 μL of 10 mM GUV was mixed with 780 μL of each concentration of PBS and 20 μL of 100 μM FITC-R8. In this case, the molar ratio of lipid/peptide (L/P) was fixed at 1000 or 850. A total of 1 mL samples was incubated at 37 °C or 2 °C, for 10 min or designated time durations in the water bath and centrifuged (4 °C, 10,500 rpm, 20 min) two times. The precipitate and 100 μL of supernatant were mixed with 200 μL of 10% TX-100 and 100% PBS to make the total volume 1 mL. The amount of translocated FITC-R8 was analyzed using the fluorescence spectrometer (Ex = 490 nm, Em = 520 nm, RF5300PC; SHIMADZU Co., Kyoto, Japan). The obtained fluorescence intensity was converted into concentration using a calibration plot of FITC-R8.

#### 4.2.8. Measurement of Cytolysis Amount of CPP

The cytolysis amount of CPP was measured by the following procedures (Appendix A). Each GUV sample treated by the Translocation assay (Section 4.2.7) was prepared, but instead of treating the final precipitate by TX-100, 20 μL of 0.05% trypsin solution was added to the precipitate and incubated again at 37 °C for 10 min. After the incubation, sample solutions were centrifuged at 10,500 rpm (for the 100 nm vesicles) and at 47,000 rpm (for 400 nm vesicles) at 4 °C for 10 min to separate the supernatant (decomposed FITC-R8 in the PBS) and precipitate (vesicles). This operation was done twice. To the precipitate, 160 μL of 10% Triton X-100 and 640 μL of 28 mM PBS (−) were added up to a total volume of 800 μL. By this operation, vesicles were disintegrated, and fluorescence measurement (Ex = 495 nm, Em = 520 nm) of internalized FITC-R8 in the vesicles was carried out using a fluorescence spectrophotometer. The obtained fluorescence intensity was converted into concentration using a calibration plot of FITC-R8.

From the fluorescence intensity obtained by fluorescence measurement and the calibration plot previously prepared, the amount of FITC-R8 by fluorescence intensity that penetrated into the vesicles was converted as the molar ratio of FITC-R8 to lipid using the following Equation (6):Amount of cytolysis (μmol/mol) = ((Amount of cytolysis FITC-R8(g))/(molecular weight of FITC-R8(g/mol)))/((N_lipo_ × N_tot_)/N_A_)(6)
where in N_lipo_ is the number of vesicles in 1 mL of suspension, N_tot_ is the number of constituent lipids per one vesicle, N_A_ is the Avogadro number.

## 5. Conclusions

Cell-penetrating peptide (CPP) can directly penetrate through the bio-membrane to the cytosol (cytolysis) and is expected to be a potent vector for DDS. Although there is general agreement that CPP cytolysis is related to dynamic membrane deformation, a distinctive process has yet to be established. Here, we introduced trypsin digestion of adsorbed CPP onto GUV to quantify the adsorption and internalization (cytolysis) separately. As a result, we found that adsorption and internalization is assumed to occur successively by CPP molecules that come into contact with membrane lipids. Adsorption is quick to saturate, while cytolysis initially proceeds depending on the amount of CPP adsorbed and then linearly by time after adsorption saturated, with the rate constant being dependent on the osmotic pressure that reflects the membrane curvature. We also found that temperature and lipid composition influences cytolysis by means of modulating lipid mobility. The electrolyte in the outer media also acted as a chemical mediator to control CPP cytolysis by following the Hoffmeister effect for the hydration of the membrane. These results confirmed the mechanism of cytolysis as temporal and local phase transfer of lipid bi-layer from Lα to Mesh_1_, which has punctured bilayer morphologies.

All these findings and analyses reminds us that phenomena in nature happen with the least amount of energy and material input with less waste in dynamic equilibrium conditions. Our knowledge of CPP cytolysis is still limited, and we have to investigate it further to improve the way to control it. We expect that a further exploration of cytolysis would open up many applications in DDS for bioactive materials to be directly internalized in the living cells based on the mechanism we have proposed with utilizing the data reported here.

## Figures and Tables

**Figure 1 ijms-21-05466-f001:**
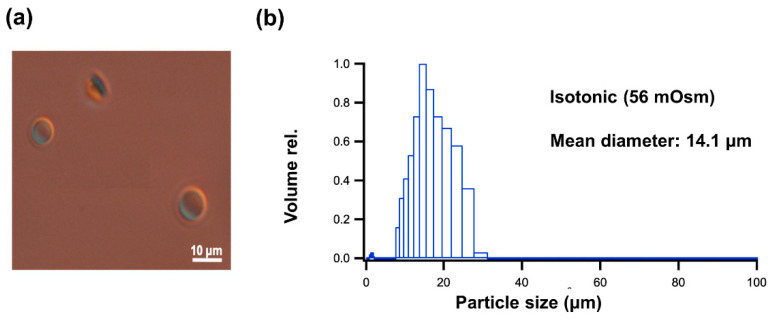
Confirmation of the GUV formation by egg yolk phosphatidylcholine (EPC). (**a**) GUV image by differential interference contrast microscope (DICM). (**b**) Particle size distribution of GUV by dynamic light scattering (DLS).

**Figure 2 ijms-21-05466-f002:**
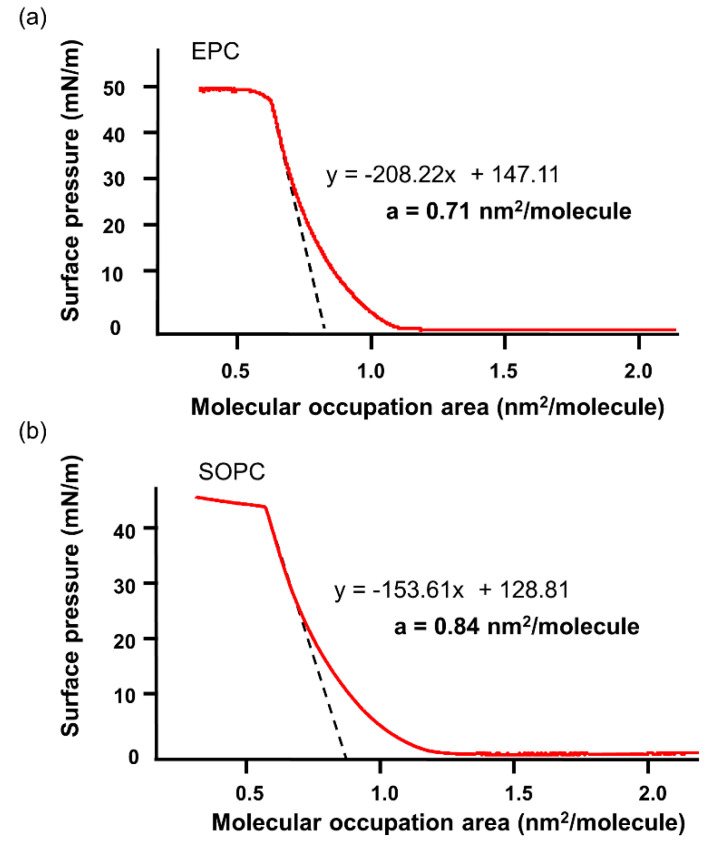
Surface pressure (π) vs. molecular occupation area (A). (37 °C, 28 mM PBS (−)), (**a**) is for EPC and (**b**) is for SOPC.

**Figure 3 ijms-21-05466-f003:**
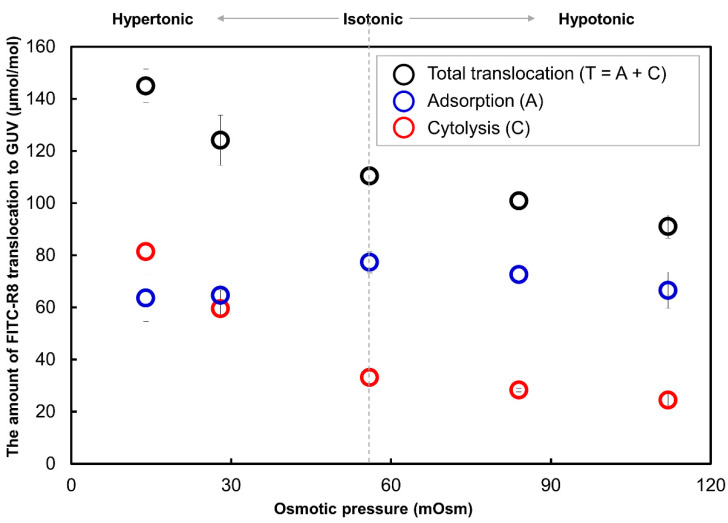
Effect of Osmotic pressure to the CPP translocation to 1-Stearoyl-2-oleoyl-phosphatidylcholine (SOPC) GUV. (FITC-R8 100 µmol; L/P = 850, incubation time 10 min, *n* = 5).

**Figure 4 ijms-21-05466-f004:**
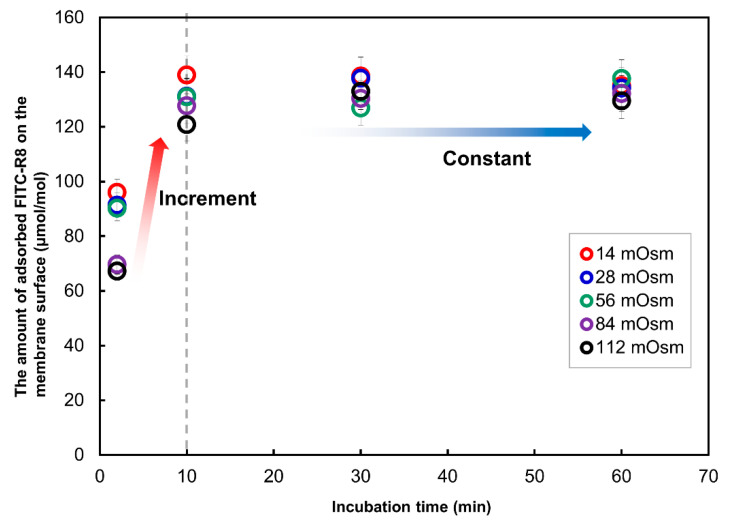
Effect of incubation time on the amount of total FITC-R8 adsorbed on the membrane surface of EPC GUV. (EPC GUV, FITC-R8 100 µM, L/P = 850, 37 °C, *n* = 5).

**Figure 5 ijms-21-05466-f005:**
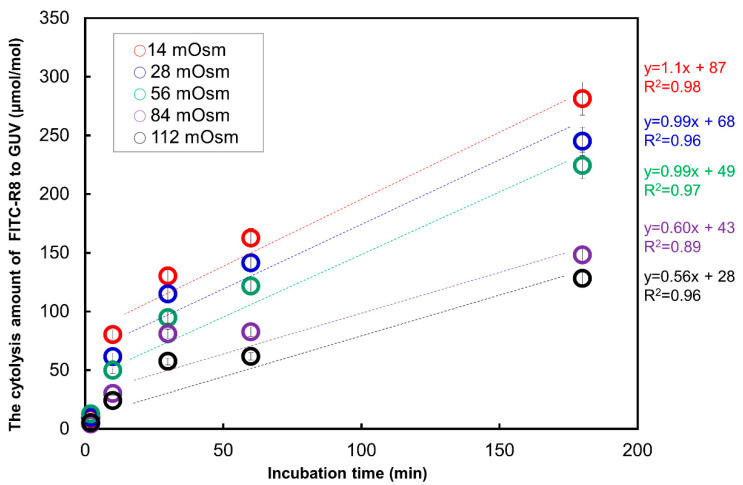
**** Effect of incubation time on the cytolysis amount FITC-R8 to EPC GUV (EPC GUV, FITC-R8 100 µM, L/P = 850, 37 °C, *n* = 5).

**Figure 6 ijms-21-05466-f006:**
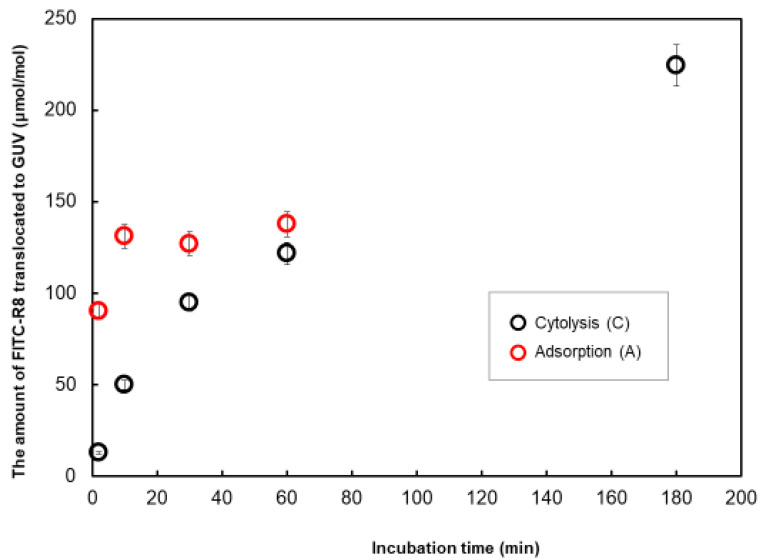
Effect of incubation time on the translocation amount of. FITC-R8 to EPC GUV under isotonic condition (EPC GUV, FITC-R8 100 µM, L/P = 850, 37 °C, *n* = 5).

**Figure 7 ijms-21-05466-f007:**
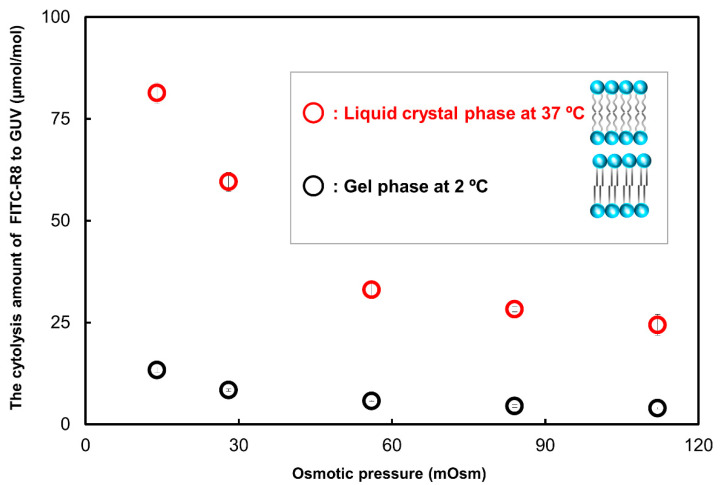
Effect of temperature on the cytolysis of FITC-R8 to SOPC GUV. (SOPC (T_c_ = 6 °C) at 37 °C or 2 °C, 100 µM FITC-R8, L/P = 850, 10 min, *n* = 5).

**Figure 8 ijms-21-05466-f008:**
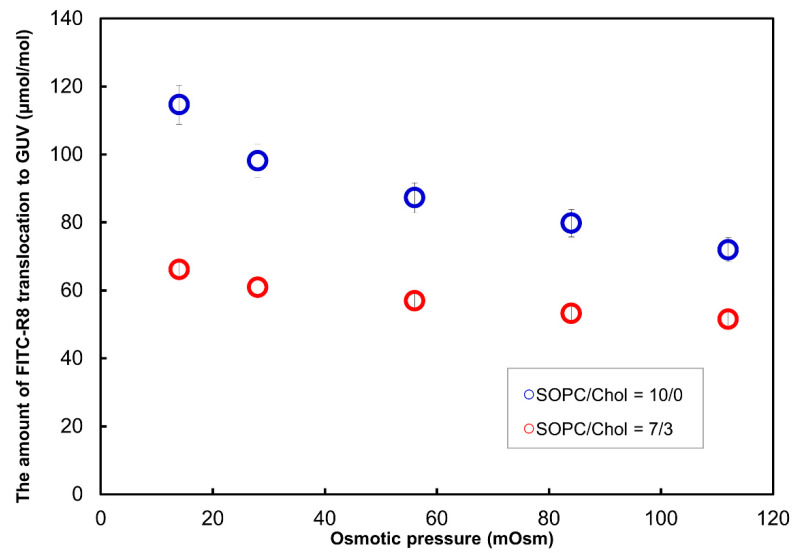
Effect of cholesterol for the CPP translocation to GUV. (SOPC/cholesterol GUV, FITC-R8 100 μM, L/P = 850, 10 min, *n* = 5, at 37 °C).

**Figure 9 ijms-21-05466-f009:**
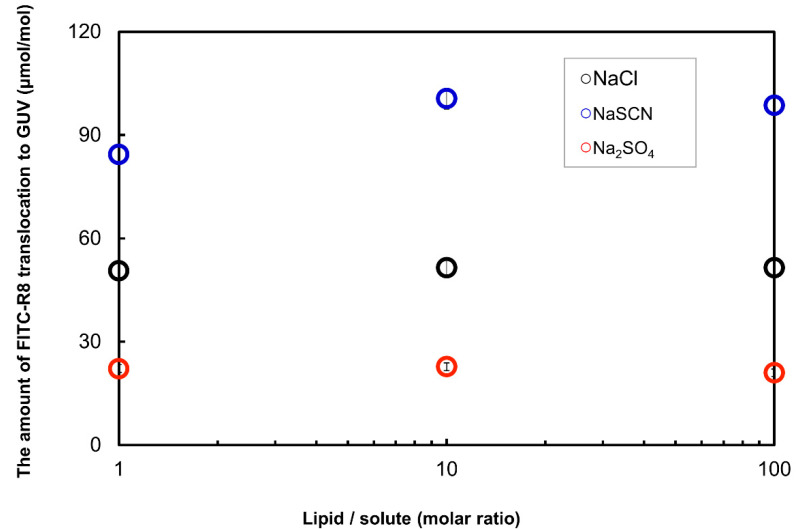
The effect of solute for the FITC-R8 translocation to SOPC GUV (SOPC/cholesterol GUV, FITC-R8 100 μM, L/P = 850, 10 min, *n* = 5, at 37 °C).

**Table 1 ijms-21-05466-t001:** Geometrical parameter of giant unilamellar vesicle (GUV).

Parameter			EPC	SOPC
d	Diameter of liposeme	nm	1.4 × 10^4^	1.4 × 10^4^
h	Thickness of bilayer	nm	4.7	4.7
a	Lipid molecular occupation area	nm^2^/molecule	0.71	0.84
N_tot_	Number of lipids per one liposome	unit	1.8 × 10^9^	1.5 × 10^9^
N_out_	Number of outer lipids per one liposome	unit	8.8 × 10^8^	8.8 × 10^8^
N_inter_	Number of internal lipids per one liposome	unit	8.8 × 10^8^	8.8 × 10^8^
N_lipo_	Number of liposome per 1 mL of liposomal suspension	unit/mL	5.8 × 10^8^	7.0 × 10^8^

**Table 2 ijms-21-05466-t002:** Membrane surface potential of SOPC GUV (L/P = 850).

PBS Concentration	mM	7	14	28
Osmotic pressure	mOsm	14	28	56
Zeta potential of GUV without FITC-R8	mV	−10.6	−10.8	−5
Zeta potential of GUV with FITC-R8	mV	−3.2	−3.9	−3.8

**Table 3 ijms-21-05466-t003:** Rate constant for cytolysis of FITC-R8 to EPC GUV.

PBS Concentration	mM	7	14	28
Osmotic pressure	mOsm	14	28	56
Rate constant *	µmol/mol EPC/min^−1^	1.1	0.99	0.96
PBS concentration	mM	7	14	28

* Rate constants were calculated from the slop in Figure 5.

**Table 4 ijms-21-05466-t004:** The amount of adsorbed FITC-R8 on the membrane surface.

Temperature (°C)	Phase	FITC-R8 μmol/SOPC mol				
		Osmotic Pressure (mOsm)				
		14	28	56	84	112
37	Liquid crystal	64	65	77	73	67
4	Gel	74	81	80	76	74

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
