# Peer review of "Key Process and Factors Controlling the Direct Translocation of Cell-Penetrating Peptide through Bio-Membrane"

_ijms, 2020, doi:10.3390/ijms21155466_

Round 1
Reviewer 1 Report
This study focuses on the internalisation process of cytolysis, which is an important issue for drug delivery system, in particular controlled delivery systems for cancer therapy.
It is concluded that adsorption and cytolysis are two independent processes. This conclusion is supported by (1) the two different time courses of adsorbed and cytolysed FITC-R8 concentration shown in Figure 4 and 5, respectively, and (2) the sparse distribution of FITC-R8 as discussed on Page 12. However, how the authors reject the hypothesis like: Adsorption is a dynamic process with a fast rate as compared to cytosis, and cytosis may be a downstream process after adsorption?
In addition, the resolution of figures is low. Some details are hard to read. This may be because of the figure quality or the compression by the paper submission system. However, it is worth to double-check to assure the figure resolution.
Some of the superscripts are not correctly written. i.e. Line 423, ‘23’ should be the superscript. Please go through the manuscript to check this issue, including the units.
Author Response
Thank you very much for your kind comments and suggestions, We are very much encouraged and inspired to make the point of this article clearer to the readers. Please kindly review the following responses.
1) It is concluded that adsorption and cytolysis are two independent processes. This conclusion is supported by (1) the two different time courses of adsorbed and cytolysed FITC-R8 concentration shown in Figures 4 and 5, respectively, and (2) the sparse distribution of FITC-R8 as discussed on Page 12. However, how the authors reject the hypothesis like Adsorption is a dynamic process with a fast rate as compared to cytosis, and cytosis may be a downstream process after adsorption?
(KS) We very much appreciate your comments. What we have intended to explain is not as “adsorption and cytolysis are two independent processes” but “cytolysis is a downstream process after adsorption” and The sparse distribution of FITC-R8 indicates that adsorption is an independent event of each FITC-R8, and adsorbed FITC-R8 is ready to proceed to the cytolysis”.
We will make the necessary corrections for the text.
2) In addition, the resolution of figures is low. Some details are hard to read. This may be because of the figure quality or the compression by the paper submission system. However, it is worth to double-check to assure the figure resolution.
(KS) Thank you very much for the comments. We are working on it to make the figure quality better.
3) Some of the superscripts are not correctly written. i.e. Line 423, ‘23’ should be the superscript. Please go through the manuscript to check this issue, including the units.
(KS) Thank you very much for the comments. We are working to correct them too.
Reviewer 2 Report
Authors have designed and obtained a Cell-Penetrating Peptide (CPP) that can directly penetrate the cytosol (Cytolysis) and expected to be a potent vector. for DDS.
Article is interesting and it is well-writen. However, I think that it will be necessary more assays in order to confirm the value of the peptide. Assays are in a very preliminar phase to publish in IJMS.
In my opinion, cell culture assays are necessary to publish (at least)
Author Response
Thank you very much for your kind comments and suggestions, We are very much encouraged and inspired to improve the introduction to make the point of this article clearer to the readers. Please kindly review the following responses.
1) Article is interesting and it is well-writen. However, I think that it will be necessary more assays in order to confirm the value of the peptide. Assays are in a very preliminar phase to publish in IJMS.
(KS) We will make clear positioning of the peptide R8 (octa-Arg) used here as a representative molecule of the CPCs in the introduction.
Futaki, the coauthor of this article, has been extensively conducting the development of cell-penetrating peptide vectors and elucidation of the internalization mechanisms mostly by cell culture studies. As explained in Ref. 1, the current flux of cell-penetrating peptide (CPP) researches is triggered by the report for the membrane translocation ability of the HIV-1 Tat protein. The importance of the domain of cationic amino acids especially Arg was confirmed and R8 and R9 are shown to be the most potent CPP as a model peptide which has the same level of internalization ability to the cultured cells as natural CPPs such as HIV-1 Tat-(48-60). Based on these study we choose R8 as CPP to investigate the mechanism of cytolysis (direct internalization) with GUV as a cell model, from a physicochemical viewpoint. As an advantage of this approach, we can avoid the interference of endocytosis and any other biological effect.
References
1) Futaki, S.; and Nakase, I. Cell-Surface Interactions on Arginine-Rich Cell-Penetrating Peptides Allow for Multiplex Modes of Internalization. Acc. Chem. Res. 2017, 50, 2449-2456.
2) Futaki, S. Oligoarginine vectors for intracellular delivery: design and cellular-uptake mechanisms. Biopolymers, 2006, 84, 241-249.
2)In my opinion, cell culture assays are necessary to publish (at least)
(KS) We have reported the cell culture assays of R8 translocation to cultured dermal fibroblast cell and erythrocyte and confirmed the translocation which are reported by Ref. 8, 9, 10, 11 in addition to Ref.1. As explained above this article is a physicochemical analysis of the cytolysis mechanism for the model system corresponding to the previous cell culture results. We will add these explanation and relating references clearly in the introduction.
References
8) Sakamoto, K.; Takino, Y.; Ogasahara, K. Method for controlling membrane permeability of a membrane permeable substance and screening method for a membrane permeable substance, Nov. 4, 2003 JP 2003-3774224(USP appl. 20050118204)
10) Sakamoto, K., Morishita, T., Aburai, K., Sakai, K., Sakai, H., Abe, M., Nakase, I. & Futaki, S. Bioinspired Mechanism for the Translocation of Peptide through the Cell Membrane, Chem. Lett. 41, 1078-1080(2012); This is a Proceedings based on the presentation made at the International Association of Colloid and Interface Scientists, Conference (IACIS2012), Sendai, Japan, May 13-18, 2012.6.
11) Sakamoto, K. The Importance of Planarity for Lipid Bilayers as Biomembrane, Adv. In Biomembranes and Lipid Self-Assembly 2016, 23, 1-23.
Round 2
Reviewer 1 Report
I have no further questions.
Reviewer 2 Report
Agree with authors clarifications included in the introduction that allow a better understanding of the work done